# Fabrication of Porous Recycled HDPE Biocomposites Foam: Effect of Rice Husk Filler Contents and Surface Treatments on the Mechanical Properties

**DOI:** 10.3390/polym12020475

**Published:** 2020-02-19

**Authors:** Farah Atiqah Abdul Azam, Nishata Royan Rajendran Royan, Nor Yuliana Yuhana, Nabilah Afiqah Mohd Radzuan, Sahrim Ahmad, Abu Bakar Sulong

**Affiliations:** 1Department of Mechanical and Manufacturing, Faculty of Engineering and Built Environment, Universiti Kebangsaan Malaysia, Bangi, Selangor Darul Ehsan 43600, Malaysia; qfaara@gmail.com (F.A.A.A.); ry_nish@yahoo.com (N.R.R.R.); yuliana@ukm.edu.my (N.Y.Y.); afiqah@ukm.edu.my (N.A.M.R.); 2School of Applied Physics, Faculty of Science and Technology, Universiti Kebangsaan Malaysia, Bangi, Selangor Darul Ehsan 43600, Malaysia; sahrim@ukm.edu.my

**Keywords:** rice husk, recycled HDPE polymer, biocomposite foam, surface treatment, mechanical properties, filler–matrix adhesion

## Abstract

In this study, a biodegradable, cheap and durable recycled high-density polyethylene (rHDPE) polymer reinforced with rice husk (RH) fibre was fabricated into a foam structure through several processes, including extrusion, internal mixing and hot pressing. The effect of filler loading on the properties of the foam and the influence of RH surface treatments on the filler–matrix adhesion and mechanical properties of the composite foam were investigated. The morphological examination shows that 50 wt.% filler content resulted in an effective dispersion of cells with the smallest cell size (58.3 µm) and the highest density (7.62 × 1011 sel/cm^3^). This small cell size benefits the mechanical properties. Results indicate that the tensile strength and the Young’s modulus of the alkali-treated RH/rHDPE composite foam are the highest amongst the treatments (10.83 MPa and 858 MPa, respectively), followed by UV/O_3_, which has shown considerable increments compared with the untreated composite. The flexural and impact tests also show the increment in strength for the composite foam after chemical treatment. Although the UV/O_3_ surface treatment has minor influence on the mechanical enhancement of the composite foam, this method may be a reliable surface treatment of the fibre-reinforced composite.

## 1. Introduction

Fabricating high-performance polymer matrix composites has been explored tremendously, and the current inventions are moving towards green technology, which has minimum emission of toxic materials and is environmentally friendly. Pertaining to this situation, numerous studies have reported on the production of natural fibre-reinforced polymer composite, particularly in polymer matrix composites [1,2]. Polymer matrix composites form the preferred group of advanced materials in structural applications, such as in aerospace, construction or automotive parts, due to their high specificity. The utilisation of recycled polymer is also one of the solutions to reduce environmental pollution. Recycled high-density polyethylene (rHDPE) is an example of a biodegradable polymer matrix that has many potential applications due to its good thermal, mechanical and dimensional stability [3,4]. Additionally, the cost of producing recycled HDPE composite is approximately 31–34% lower than that of using virgin HDPE. The main reason is that the plastic waste like post-consumer HDPE is sold for less than half the price, but the mechanical performance of recycled HDPE composite was as good as virgin HDPE composite [5]. Thus, the use of recycled HDPE not only minimises solid waste disposal, but also reduces the manufacturing cost of HDPE-based products for household users or the automotive industry.

Furthermore, as widely reported, the usage of natural fibre from plants, such as kenaf [6], hemp, jute [7,8], bamboo, flax and rice husk (RH) [9,10,11], as reinforcement in a polymer matrix improves the physical and mechanical properties of the polymer [12,13,14]. Wood plastic composite (WPC) has been widely investigated by current researchers. The usage of plant-based natural fibre as fibre reinforced polymer (FRP) able to produce lighter polymer composites with a better strength and rigidity for many applications [15]. In addition, the usage of recycled polymer and natural fibre promotes the biodegradable process, and economically competitive materials can be produced [11]. RH is an efficient agro-filler in the production of lightweight polymer-based composites. Many studies on RH-reinforced composites have been conducted [11,16,17,18,19]. Some factors that encourage the usage of RH in the composite material industry are its availability, low density (90–150 kg/m^3^), durability and weather resistance [20]. Phan et al. [21] highlighted on the improvements in combustibility properties of RH reinforced polyurethane (PU) that has overcome the disadvantages of highly flammable PU. However, the existence of dirt, wax and lignin on the surface of RH and the hydrophilic properties of hemicelluloses are the major factors that contribute to the poor adhesion between the RH and the polymer matrix. The surface modification of RH through chemical treatments, such as alkali, citric acid, anhydride and UV/O_3_ surface treatments, are commonly conducted to enhance the filler–matrix adhesion [22,23,24,25]. The surface treatment of RH by using UV/O_3_ surface treatment has been introduced several years ago. This method is cheaper and environmentally friendly compared with other chemical treatments [26,27,28]. Our previous study has demonstrated that this new alternative treatment is proven to have a great rice husk (RH) surface to improve the adhesion between the hydrophilic RH fibre and the hydrophobic rHDPE polymer matrix [18]. In addition, UV/O_3_ treatment helps in removing surface impurities and avoiding the degradation of organic materials, like carbonate and silica [28,29]. This treatment also causes the removal of wax and degradation of lignin without disrupting the hemicellulose and cellulose of the fibre. Ozonolysis is cheap and reduces the damage to the natural environment [26]. 

Nevertheless, the usage of treated fibre as reinforcement for application in polymer composite foam is still being developed. The fabrication of foam-based WPC by using polymer matrix and natural fibre as a filler has advantage in density and material costs and may increase the strength of the composite foam simultaneously. The weight of the polymer composite with RH as filler can be further reduced if a foam agent is introduced. Natural fibre/filler can be a good nucleating agent of the foam. However, introducing fibre as reinforcements may also cause nonuniform cell size, irregular cell nucleation or a low cell density foam. This condition may deteriorate the physical and mechanical properties of the composite foam and is the challenge of using natural fibre as a filler. Some factors that may affect the size and distribution of cell size and cell nucleation are the molecular weight of the polymer foam, the chemical or physical blowing agents used and the amount of filler or reinforcement. This study highlights the filler content and surface treatment effect of the porous structure of rHDPE biocomposite. Above all, researchers are giving great consideration to fibre-reinforced polymer composite foam rather than the solid polymer composite because the development of polymer-based foam composite can overcome the weaknesses of the solid composite due to its density, cost and impact strength. Foam technology consists of a different series of processing methods to produce WPC parts by creating large quantities of foam by foam agents by using a physical or a chemical foam agent [14,30,31]. 

Many studies have reported the surface modification of the fibre and the development of the fibre-reinforced solid polymer composite [22,32]. Previously, we have reported on the advantages of UV/O_3_ surface treatment to the rice husk filler of rHDPE solid biocomposite and its mechanical properties [28]. However, to the authors’ knowledge, none studies have been performed on the effect of fibre reinforcement and surface treatment on the porous structure of rHDPE polymer biocomposite foam, especially after UV/O_3_ surface treatment. Relatively, this study has focused on the effect of RH fibre content on the foam structure and the influence of untreated and surface-treated RH fibre on the morphology and mechanical properties (tensile, flexural and impact) of the produced rHDPE foam.

## 2. Materials and Methods 

### 2.1. Materials

rHDPE acts as a matrix in this biocomposite. The rHDPE has a density of 923 kg/m^3^ and a melt flow index of 0.072 g per 10 min at 190 °C. RH natural fibre with a mesh size of 212 µm was used as filler and obtained from Din Xing (M) Sdn. Bhd. (Guar Chempedak, Kedah, Malaysia). The coupling agent used in this study was maleic anhydride-grafted polyethylene (MAPE) supplied by Muda Cemerlang Sdn. Bhd (Rawang, Selangor, Malaysia), which also supplied rHDPE. Additionally, the exothermic chemical blowing agent azodicarbonamide (ADC) was supplied by Sigma Aldrich (Petaling Jaya, Selangor, Malaysia).

### 2.2. Sample Preparation

#### 2.2.1. Surface Treatment of RH Fibre

Ultraviolet (UV/O_3_), alkali and acid treatments were used in this study. The UV/O_3_ treatment was conducted using a self-made in-house UV/O_3_ system. In the chamber, RH was exposed to 10 L/min ozone flow rate for 30 min. All parameters in this study were based on previous studies [26] designated from the United States Patent (Wydeven), Patent No: US 6,555,835 B1 [33]. Initially, the sample was placed in a conical flask, which had ozone gas input and output flow channels, and in a reaction chamber with UV lamps (254 nm). The photosensitised oxidation occurred inside the reaction chamber. For the alkali treatment, RH was soaked in 0.5 M NaOH solution for 2 h and stirred using a magnetic stirrer. The alkaline residue on the RH surface was removed using distilled water until a pH of 7 was reached. Finally, the treated RH was placed into a vacuum oven at 90 °C and left to dry overnight. This method was adapted from a study by Garcia et al. [23]. Meanwhile, a high-concentration acid solution was prepared for acid treatment by using sulphuric and nitric acids at a ratio of 3:1 (*v/v*). Then, RH was soaked in the dilute acid solution and refluxed under an oxidation environment. The oxidised RH was then washed with distilled water until a pH of 7 was reached. Lastly, the treated RH was dried in an oven maintained at 90 °C for a day.

#### 2.2.2. Fabrication of the Composite Foam

Prior to mixing, RH was washed with distilled water to remove the impurities on its surface and oven-dried at 80 °C for 24 h to reduce the moisture content. Then, polymer composites containing 50 wt.% of untreated and treated RH were prepared. The dried RH fibres and rHDPE were mixed constantly with 4% MAPE coupling agent and compounded together by using a laboratory-scale counter-rotating twin-screw extruder. The screw rotation rate was set at 30 rpm. The barrel temperatures applied from the feed zone towards the die zones were 180, 190, 200 and 190 °C. The MAPE content was selected on the basis of a pilot study on the effect of MAPE content on the mechanical properties of RH/rHDPE composites, wherein 4 wt.% MAPE yielded the best results [10]. Then, the resulting mixture was transferred into an internal mixer. The RH fibre, rHDPE, MAPE and blowing agent were compounded at a foaming decomposition temperature of 120 °C, rotation speed of 60 rpm and mixing time of 10 min. All mixing methods and parameters were referred to studies conducted by Hemmasi et al. [29] and Matuana et al. [34]. The composition of the matrix and filler materials was set at 50 g. 

In this study, approximately 6 wt.% ADC powder was used as an exothermic blowing agent. ZnO was used as an activator to reduce heat release from the ADC during compaction. The ZnO quantity was kept constant at 1.5% by weight in all formulations. This quantity was selected on the basis of the previous studies, which reported that this action reduced the optimum ADC heat release. After the mixture followed the curve of the match for approximately 2 min, ADC and ZnO were added. Then, the mixture was left for 10 min and milled using a milling machine (LP50, LABTECH Engineering Company, LTD. Machine Samutprakarn, Thailand and incorporated into a mould with an area of 14 × 14 mm and thickness of 3 mm to undergo the hot and cold pressing. The temperature and pressure were set at 180 °C and 6.9 MPa, respectively. After the incubation period (10–20 min), the specimen was cooled for approximately 3 min. The pressure on the samples was maintained at 1000 psi. Then, the pressure was removed, and the composite sample was left to expand [29]. Figure 1 shows the process of producing the RH/rHDPE composite foam. 

### 2.3. Characterisation

The micrographs of the RH/rHDPE composite foam, which consisted of untreated and treated RH, were also obtained using a field emission scanning electron microscopy (SEM) (model: SUPRA 55VPSEM). The cell size and density of the untreated RH of RH/rHDPE composite foam sample with different percentages of RH (10, 30 and 50 wt.%) were measured and calculated using Equations (1) and (2) on the basis of the composite morphology. Subsequently, mechanical characterisation was performed on the composite foam with the optimum RH content. Tensile, flexural and notch impact tests were conducted to examine the mechanical properties of the prepared composite. The specimens were cut in accordance with the ASTM D 256-10 specifications [35]. 

Tensile strength was determined using a Materials Testing Machine (model: M350-10CT) with a 5000 N load and a speed of 5 mm/min according to the ASTM D 638-03 standards [36]. The flexural strength of the sample was studied using three-point flexural tests. The sample was cut into a rectangular shape (127 × 12.7 × 3 mm) in accordance with the ASTM D 790-03 standards and examined with 5000 N load and a speed of 5 mm/min using a flexural testing machine model (Testometric M350-10CT) [37]. The modulus of elasticity was determined directly from the machine. Meanwhile, an impact test was carried out in accordance with the ASTM D256-10 standards by using the Izod type (Ray-Ran Universal Pendulum Impact System) with a pendulum velocity of 3.46 m/s and a weight load of 0.452 kg. A rectangular-shaped sample with the dimensions 63.5 × 3.0 × 12.7 mm was used in this test.

## 3. Results and Discussion

### 3.1. Cell Size, Density and Morphology of the HR/rHDPE Composite Foam

The evaluation of foam morphology by using cell size and cell density is fundamental, especially for the interpretation of the mechanical properties. The cell size is characterised by measuring the cell diameter and determined by analysing the SEM micrographs. The effect of RH fibres addition on the average cell size and cell density of the RH/rHDPE-based composite samples with constant ADC content (6 wt.%) is shown in Table 1. Notably, the 50 wt.% RH/rHDPE composite sample has the minimum cell size (58.3 µm) and the highest density (7.62 × 1011 sel/cm^3^). As the RH fibre content increases, decreased average cell size and increased cell density are observed. This is for the reason that, the high RH content leads to an increase in the viscosity of the matrix, which inhibits cell growth during foam formation [34,38]. 

During foaming decomposition temperature at 120 °C, good wetting between phases promotes void nucleation and facilitates the nucleation of particles. The void nucleation increases the cell density and decreases the distribution of cell size [39]. At constant amount of blowing agents (6 wt.% ADC), the gas produced for the foam formation is constant. In other words, as the fibre loading increases, a large number of cells limit the cell growth by using the available gas. As a result, the cell size decreases, and its spread is constricted. Therefore, 10 wt.% RH filler creates the highest cell sizes, whereas 50 wt.% RH results in the lowest cell sizes in foam composites. 

As expected, the composite foam with 50 wt.% RH clearly shows the finest cell structures (≤64 µm) amongst the compositions with uniform cell size dispersion. The high content of RH filler increases the viscosity, which denotes high resistance to foaming and consequently causes small cell size. In addition, as mentioned previously, the high amount of fibre leads to numerous numbers of nucleation sites through heterogeneous mechanisms. Heterogenous nucleation occurs at the interface of solid fibre and liquid polymer, that reduces the surface tension and results in the increase in system free energy. This condition leads to small cell size and high density that trigger an improvement in the mechanical properties of the polymer composite [14].

The amount of void in the RH–rHDPE foam was represented as void fraction (V_f_) and cell density (N_f_), which is the number of cells per unit volume, and calculated using Equations (1) and (2). The average cell size (d) was calculated using Equation (3) [29,34].
(1)$$V_{f}=1-\;\frac{\rho_{f}}{\rho_{f}}$$
(2)$$N_{f}\;=\;{\left(\frac{{\mathrm{nM}}^{2}}{A}\right)}^{\frac{3}{2}}.\;\left(\frac{1}{{1-\;V}_{f}}\right)$$
(3)$$d\;=\;{\left(\frac{{6\rho V}_{f}}{{\pi N}_{0}\left({1-V}_{f}\right)}\right)}^{\frac{1}{3}}$$
where ρ is the density of solid composite, ρ_f_ is the density of foam composite, n is the number of cells in the micrograph, A is the area of micrograph in cm^2^, M is the enlargement factor, and d represents the average cell size.

The mechanism of foaming process initiated by the nucleation of cells. RH fibres function as nucleation agents, which promote hezterogeneous nucleation at the solid-liquid interfaces. It is happened when the decrease of surface tension at the interfaces of solid fiber and the liquid polymer caused an increase in the free energy of the system [29]. Throughout mixing, the chemical reaction occurs, the polymer in this solution becomes supersaturated, and when the temperature reaches foaming decomposition temperature, heterogenous nucleation occurs. Then, cells start to grow, and the number of cells increases. As the cells grow enormously, and the foam volume expands continuously, the foam system becomes unstable. The blowing agent controlled the foam formation from the liquid and it is slowly stabilised after cooling then solidified [40]. 

Aside from foam density, the foam properties are greatly affected by the geometry of the cell (either open or closed cells) and the distribution of cell size and its uniformity. The cell size and density are used to evaluate the morphology of cells. The SEM micrographs of RH/rHDPE-based composite with different RH contents (10, 30 and 50 wt.%) are shown in Figure 2a–c. All composites show the closed cell structure with of macrocellular foam (cell dimension > 100 µm) for 10 wt.% RH and microcellular foam (cell dimension = 1–100 µm) for 30 wt.% RH and 50 wt.% RH. Many studies have reported that polymer composite with microcellular foam has superior impact strength [38,41], toughness [42], stiffness-to-weight ratio [43] and lower material weight and cost compared with the conventional foam or unfoamed polymers. In general, microcellular plastics with cell size <10 µm and cell density >10^9^ cells/cm^3^ have better mechanical properties compared with the conventional foam products [33]. Guo et al. [30] stated that the mechanical properties of the composite is proportional to the cell density. Notably, the of RH fibre in the polymer matrix has enhanced the morphology of the composite foam cells (Figure 2).

### 3.2. Mechanical Properties: Effects of the Interfacial Modification of Fibre

#### 3.2.1. Tensile Test

The tensile strength and the Young’s modulus of the untreated and treated RH/rHDPE composite foam containing 50 wt.% RH (optimum loading of RH) is shown in Figure 3. Remarkably, the alkaline-treated RH/rHDPE composite has shown the highest tensile strength which is 10.83 MPa (44.4–71.9% higher) compared with the other treatments. A similar pattern is observed for the Young’s modulus, wherein the alkali-treated RH/rHDPE composite foam has the highest Young’s modulus (858 MPa), followed by the UV/O_3_, acid-treated and untreated composite foams. Evidently, surface treatment enhances the tensile strength of the composite. Based on the functional group analysis as reported in the previous study by Royan et al. [28], the photosensitised oxidation occurs during UV treatment and causes a postulated chemical reaction. This oxidation generates the ozone molecules, which consist of carboxyl, hydroxyl, ester and carbonyl functional groups on the surface of the treated RH. This phenomenon results in a rough surface of the fibre that promotes great adhesion between RH filler and rHDPE polymer matrix, thereby improving the mechanical properties. The good foamability due to the surface treatment has induced favourable foam structures, which enhance the ductility of materials. Remarkably, the tensile strength of the composite is even better when using UV/O_3_ treatment than acid treatment. As compared to untreated RH composite, the tensile strength has increases about 20.5%. This result indicates that instead of using harsh chemical treatments (like acid), UV/O_3_ can be used to improve the stiffness of this composite foam by improving the interlocking of the fibre–matrix interface. Some other observations in the previous studies also prove the reliability of UV/O_3_ as a surface treatment to enhance the filler–matrix adhesion of composite [26,27]. Moreover, this method is cheap and environmentally friendly.

A distinct observation is made when RH/rHDPE composite foam is compared with the solid composite by using the UV/O_3_ treatment. Royan et al. [28]. reported that UV/O_3_ treatment of RH for rHDPE solid composite exhibits the highest tensile strength (18.5 MPa) compared with acid or alkali chemical treatment. Interestingly, the Young’s modulus of UV/O_3_-treated composite foam in this study (530 MPa) is much greater than that of UV/O_3_-treated RH/rHDPE solid composite (~200 MPa). This experimental result indicates that the composite foam with a porous structure can have higher strength compared with the solid composite. This report is also supported by Matuana et al. [38] who reported that the foam morphology is affected by the surface treatment and causes higher stiffness of foamed composites with 56% of void fraction compared with the unfoamed polyvinyl chloride. Another possible reason that explains the high elasticity of this composite foam is because of higher RH content in this study compared with the previous study, which has used only 30 wt.% RH [28]. Tong et al. [10] explained that tensile strength is strongly dependent on the content of fibre loading, whereas increasing the RH filler tends to increase the rigidity and stiffness of the composite up to a certain amount. The rigidity of the composites is much better compared with the unreinforced polymer due to the good stress propagation inside the interfacial bonding of fibre and polymer matrix. This result has been explained by Rana et al. [8] and Ismail et al. [25] in their work. 

#### 3.2.2. Flexural Test

In flexural tests, the extension (convex side) and the compression (concave side) stresses have occurred on both sides of the composite, as opposed to the tensile stress which simply involves one stress in a uniaxial direction. Thus, the homogeneity of filler dispersion in the composite and filler–matrix interaction plays an important role in achieving the good transmission of stress in the composite. Good flexural properties can be attained by having a good dispersion and enhanced filler–matrix interfacial bonding [12]. Figure 4 shows the flexural strength and elastic modulus of the composite foam at 50 wt.% RH loading of untreated RH/rHDPE and UV/O_3_-, and alkaline- and acid-treated RH/rHDPE. Surprisingly, the untreated and chemically treated RH in composite foam by using acid and alkali seems to exhibit not much difference in flexural strength with approximately 28–30 MPa. The elastic moduli of the untreated and the treated composites also display an almost negligible effect (Figure 4b). The untreated composite foam shows an elastic modulus of 2215 MPa. Although the increment in strength is trivial with only 1.7–3.5% compared with the untreated composite, the chemical treatment has a positive effect on the flexural properties. According to Nourbakhsh et al. [17] the high quantity of RH may also be the result of filler agglomeration, which becomes an obstacle to optimum flexural strength.

Previous research by Royan et al. [28] proved that the UV/O_3_-treated filler can improve the mechanical properties of the solid-reinforced rHDPE composite with better tensile and flexural properties than acid or alkali treatment. Remarkably, UV treatment, which is considered as a mild treatment compared with acid and alkali treatments, has minimised the degradation of the particles and resulted in the optimum mechanical properties. Contradicting this study, the flexural strength of UV/O_3_-treated composite even dropped to 19.5 MPa compared with the untreated composite (29 MPa). This result is possibly related to the distinct structure of solid and foam composite with porous structure [44]. The propagation of stress that occurs within the composite foam acts differently than the uniaxial tensile test, which occurs only in one axis direction. In flexural tests, the existence of many microcells in the structure of composite foam may easily restrict the transmission of extension and compression stresses that occur simultaneously on both sides of the composite and further induce the propagation of cracks. Insufficient filler–matrix interaction after the UV/O_3_-treatment consequently reduces further ductility of the composite foam, which may be the possible reason for the reduction of elastic modulus and flexural strength [17].

#### 3.2.3. Impact Strength

The impact strength of the RH/rHDPE composite foam on 50% RH loading of untreated RH and UV/O_3_-, alkali- and acid-treated RH was evaluated. The noticeable superior result of impact strength for composite foam treated by alkali and acid treatments is shown in Figure 5. A similar observation is previously made for the elastic modulus. This result is in accordance with the work by Chong et al. [45] which reported that chemical treatment increases the roughness of the RH surface, and fibril fibrillation caused by NaOH treatment increases the surface area. This result favours the mechanical interlocking between the fibre and matrix. In a previous study, the author has reported that the alkali and acid treatments used are able to successfully remove the surface impurities, such as wax, hemicellulose and lignin on the RH surface based on the functional group analysis [28]. An improved interaction between RH fibres and the matrix foam after acid or alkali surface treatment is due to the esterification mechanism. This condition causes a good propagation of stress amongst the filler and matrix. This situation leads to the increment in the impact strength of alkali- and acid-treated RH for the RH/rHDPE composite. The stiffness of the composite is improved after the chemical treatment due to the high force transfer at the fibre–matrix interface in the composite during the impact test [28]. Impact properties are also highly related to the dispersion of fibre reinforcement. A poor dispersion of filler in the untreated composite foam may cause several filler–filler interactions. If this filler–filler interaction is greater than the filler–matrix interaction, the crack sensitivity of the composite foam may increase [46,47].

Nevertheless, the composite foam treated with UV/O_3_ shows an almost negligible increment of impact strength (approximately 1.51%) compared with the untreated RH composite. Notably, the low impact strength of the composite with UV/O_3_-treated RH is due to insufficient filler–matrix interactions, which cause a low capability to absorb a sudden application of load. Although UV/O_3_ treatment removes the impurities on the surface of the RH filler, microcells may still exist between the filler–matrix interfaces of the foam structure and can reduce filler–matrix adhesion [13,48]. This condition subsequently triggers several dispersions of cracks in the composite foam and results in low impact strength. This result has been explained by Chen et al. [46] and Yang et al. [49] who have shown that the existence of those spaces induces microcracks, resulting in microcrack propagation during the impact. Thus, these results demonstrate that acid and alkali surface treatments seem to produce good filler–matrix interactions without being affected by the porous structure of the composite, thereby further enhancing the impact strength. However, mild treatments, like UV/O_3_, have high sensitivity to cracks during impact and low adhesion at the filler matrix interface.

### 3.3. Morphology of RH/RHDPE Composites with Untreated and Surface-Treated RH

The morphology of the RH/rHDPE composite with untreated and surface-treated RH was examined using SEM. Figure 6b–d show a coarse surface, where the rough surface helps improve the mechanical properties of the RH fibre and the polymeric matrix. Hemmasi et al. [29] stated that the mechanical properties of the composite foam are closely related to the foam morphology because the foam properties may depend on the geometry of cells (either open or closed cells) and the distributions and uniformity of cells in the WPC foams. An open structure and nonuniform dispersion of cells may contribute to the low mechanical properties of the composite. Figure 6b,d clearly show that the composites have a closed cell morphology with nearly uniform sizes and shapes. Composite foam with surface-treated RH causes a good dispersion of RH fibre in the polymer matrix, and the homogeneity of cells cause several gaps to be filled with foam.

However, Figure 6c displays that alkali treatment has distorted the structure of the foam. The alkali treatment seems to cause a very strong adhesion of fibre to the polymer matrix, thus increasing the viscosity and developing the restriction in the formation of cells during foaming. The high adhesion of fibre–matrix and high viscosity may result to undissolved gas bubbles during nucleation, causing the irregular shapes of voids [29,34,50]. High viscosity and small cell size may enhance the mechanical properties. However, although the alkali treatment of RH has caused fluctuations in the structure of composite foam, the strong adhesion of RH fibre–rHDPE matrix has concealed the voids and reduced the number of cells in the composite foam (Figure 6c). The alkali treated RH composite have a nonuniform cell sizes has not produce a good pore structure. The composite becomes more compacted and this phenomenon may be the reason for the higher strength of alkali-treated RH composite foam compared with other composite foams. 

## 4. Conclusions

The effect of RH filler loading on the morphology and surface treatment of the composite foam on the mechanical properties of the RH/HDPE composite foam is investigated. The experimental results show that the increment of the fibre content in the composite up to 50 wt.% produces the lowest cell size (58.3 µm) and the highest density (7.62 × 1011 sel/cm^3^), thus enhancing the suitability for mechanical evaluation. Results also imply that besides the cell size and the filler distribution, the chemical treatment of fibre may affect the mechanical properties of foam-based composite. Even though alkali treatment displays the highest tensile (10.83 MPa) strength, the morphology shows that the foam structure of the composite is distorted and that the cells’ shapes and sizes are not desirable. Results also show that the UV/O_3_ surface treatment of RH fibre increases the Young’s modulus and tensile strength of the rHDPE composite foam up to 20.5% but does not have an increment effect on the flexural and impact properties of rHDPE foam. However, this treatment still has potential for use in the surface modification of RH as reinforcement for composite foam, replacing other harsh chemical treatments. 

## Figures and Tables

**Figure 1 polymers-12-00475-f001:**
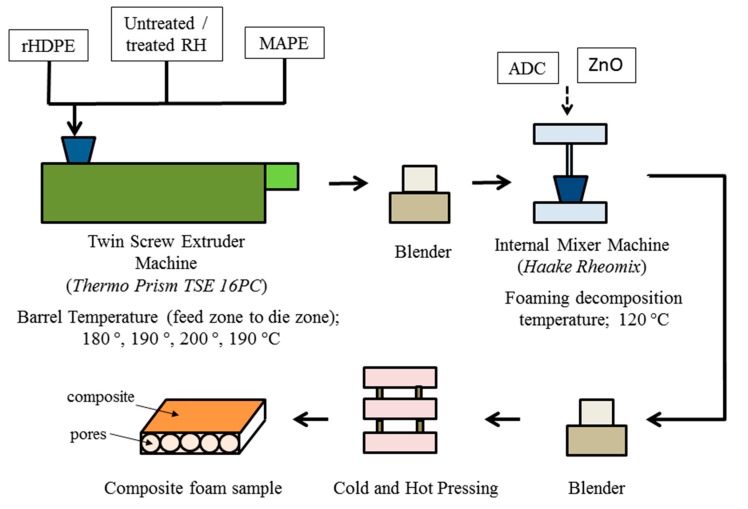
Fabrication of rice husk (RH)/recycled high-density polyethylene (rHDPE) polymer composite foam.

**Figure 2 polymers-12-00475-f002:**
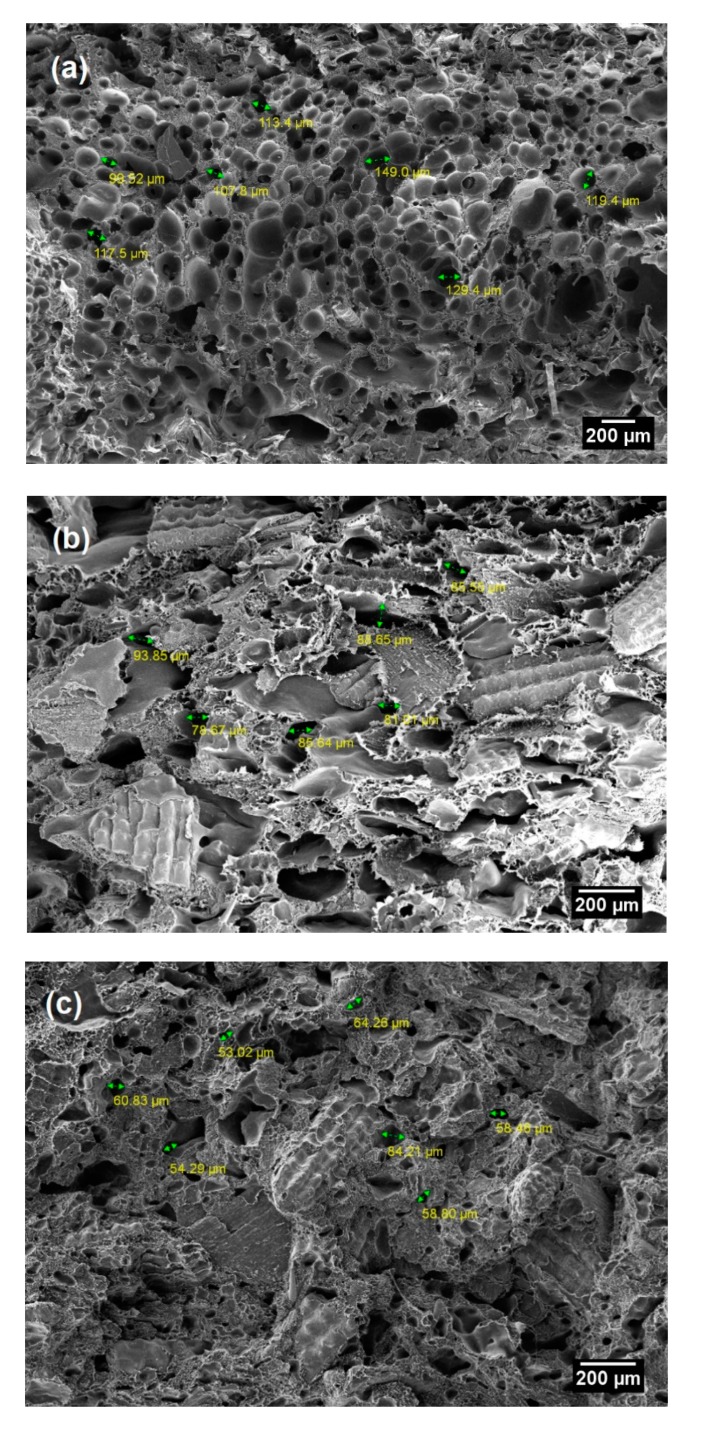
SEM micrographs of RH/rHDPE composite foam with different contents of untreated rice husk (RH) filler: (**a**) 10, (**b**) 30 and (**c**) 50 wt.%.

**Figure 3 polymers-12-00475-f003:**
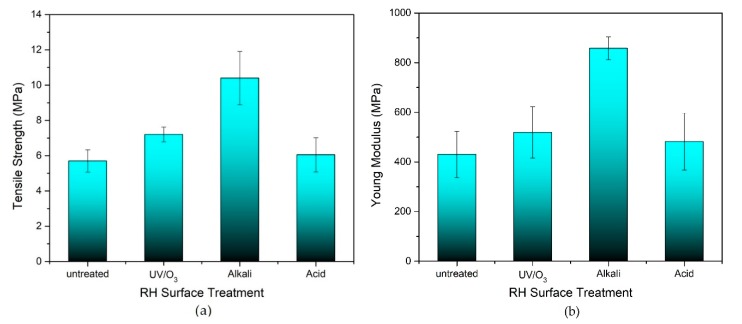
(**a**) Tensile strength and (**b**) Young’s modulus of untreated, alkali- and UV-treated RH with 50 wt.% RH.

**Figure 4 polymers-12-00475-f004:**
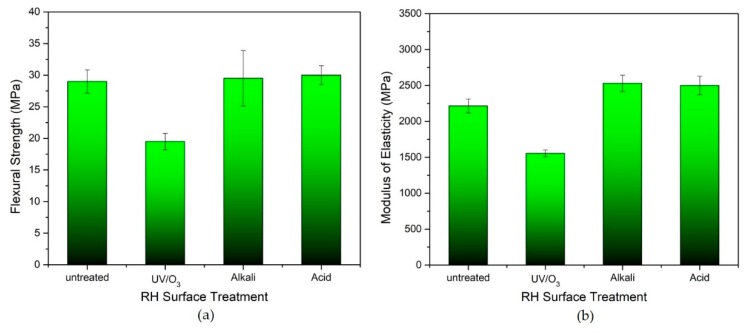
(**a**) Flexural strength and (**b**) modulus of elasticity of untreated RH, UV-, alkali- and acid-treated RH with 50 wt.% RH loading.

**Figure 5 polymers-12-00475-f005:**
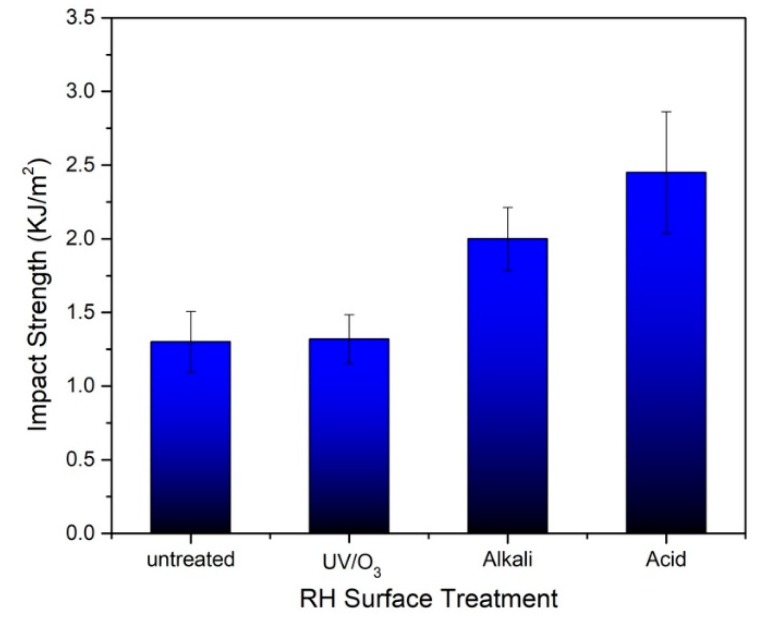
Impact strength of untreated, UV-, alkali- and acid-treated RH with 50 wt.% RH loading.

**Figure 6 polymers-12-00475-f006:**
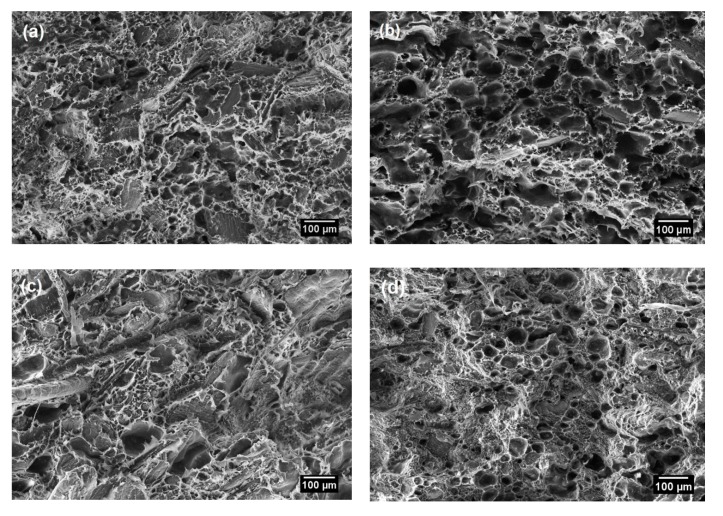
SEM morphology of (**a**) untreated, (**b**) UV/O_3_-, (**c**) alkali- and (**d**) acid-treated RH/rHDPE composite foam of 50 wt.% rice husk filler.

**Table 1 polymers-12-00475-t001:** Effect of rice husk loading on the cell size and cell density of RH/rHDPE composite foam.

Composite Sample (RH/rHDPE/ADC)	Cell Size (µm)	Cell Density (cell/cm^3^)
10 wt.% RH	122.8 ± 4.1	4.89 × 10^11^
30 wt.% RH	85.6 ± 2.7	5.94 × 10^11^
50 wt.% RH	58.3 ± 3.2	7.62 × 10^11^

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
