# Peer review of "Fabrication of Porous Recycled HDPE Biocomposites Foam: Effect of Rice Husk Filler Contents and Surface Treatments on the Mechanical Properties"

_polymers, 2020, doi:10.3390/polym12020475_

Round 1
Reviewer 1 Report
Revision of the article ''Fabrication of Porous Recycled HDPE Biocomposites Foam: Effect of Rice Husk Filler Contents and Surface Treatments on
the Mechanical Properties''.
The authors improved the work significantly. Enough to print in the Polymers journal. Both Introduction and the research part have been supplemented and improved. The doubts of the reviewers were clarified. I accept the article for printing.
The reduction of HDPE production costs with rice husk has not been fully explained (line 41-42). This is only a general description of the cost reduction of foam with the addition of recyclate.
The purpose of chemical treatment of RH husk was explained.
The authors claim that the method chosen by them is cheaper than others [21-22]. Ozonolysis is cheap and reduces the damage to the natural environment [20]. The dependence of density on cell strength and shape was discussed.
Author Response
Dear Reviewer,
Thank you for all your supportive comments. We have made the changes in the manuscript accordingly. Your comment as the following has been improved in the manuscript.
Point 1: The reduction of HDPE production cost has been not fully explained. This is only a general description of the cost reduction of foam with the addition of recyclate.
Please refer to the line (41-45) for the amendment that has been made.
Additionally, the cost of producing recycled HDPE composite is approximately 31%–34% lower than that of using virgin HDPE. The main reason is that the plastic waste like post-consumer HDPE is sold for less than half the price, but the mechanical performance of recycled HDPE composite was as good as virgin HDPE composite[5]. Thus, the use of recycled HDPE not only minimises solid waste disposal, but also reduces the manufacturing cost of HDPE-based products for household users or the automotive industry.
All the imposed changes are in yellow colors in the manuscript.
Thank you.
Regards.
Reviewer 2 Report
The authors have properly addressed the comments made in their previous submission.
At this point, I have nothing else to add, so I recommend submission.
Author Response
Dear Reviewer,
Thank you for your useful comments.
This manuscript is a resubmission of an earlier submission. The following is a list of the peer review reports and author responses from that submission.
Round 1
Reviewer 1 Report
In present manuscript, a kind of composite foams were fabricated using rice husk and recycled HPDE as raw materials. The authors focused on the effect of the rice husk content and the surface treatment methods on the structure and mechanical properties of the composite foams. However, the manuscript in current form was more like an experiment report other than a scientific paper, and it is very difficult to find any highlight. No new mechanisms and methods were reported here, which is of little reference value to readers. Upon above justification, this manuscript is not acceptable for publication in Polymers.
There are some details need to be noticed as follows:
1. Line 175-176, page 5. The authors should give the detail explanation to the viewpoint that “as the rice husk content increases, the melting point of matrix increases”.
2. Line 181-184, page 5. How to prove the function of the heterogeneous nucleation?
3. Table1, what is the reliability of the cell size data? What is the statistical error?
4. What are the mechanisms of the surface modifications using UV, alkali and acid treatments? What happens to the chemical groups on the surface of RH? If there is no analysis of modification mechanism, how to explain its influence on the properties of foams.
5. Line 316-320, page 9. It is difficult to find how to supports the results of the mechanical properties of foams by the SEM images in Fig 6.
Reviewer 2 Report
I have received the article entitled: “A Simple Approach to Fabricated Rice Husk Reinforced Recycled HDPE composite foam and its Mechanical Properties” for publication at MDPI Polymers. The article proposes to evaluate a polymer composite foam from recycled high density “polypropylene” (rHDPE) reinforced by rice husk filler through extrusion, internal mixing, and hot pressing.
The first issue that the authors must address is that, if they are trying to publish in MDPI Polymers, they at least should take care of writing properly the names of the used polymers. Even though during the whole manuscript the authors talk about polyethylene, in accordance with the title, the fact that you made this mistake in your abstract attracted my attention, in general an intensive English edition is needed, for which authors can address the Editorial services provided by MDPI (https://www.mdpi.com/authors/english) or by any other provider of your choice.
Whereas I performed a plagiarism checker, and the results can be considered acceptable, I would ask the authors to clearly define what is the relevance of this work, as after reading
10.15376/biores.10.4.6872-6885
10.1371/journal.pone.0197345 May
Journal of Mechanical Engineering Vol SI 3 (2), 13-22, 2017
10.1155/2014/938961
It appears to me that you are over-exploding the study of rice-husk reinforced rHDPE, while after reading your results it appears that it is not the case. Moreover, I find the descriptions in Materials sections filled with wordiness. Please make a deeper analysis of the kinetics of foam-forming, as this is actually the new feature of your HDPE-rice husk composites.
The work is sustained by good and interesting results; however, as I said before, the title must be changed, as at first glance it sounds like more of the same. The introduction should be more focused on the presentation of the problems that you have addressed and want to solve with this work, with a particular emphasis on your own experience, which in addition will serve to clarify the innovations made regarding your previous works. The methodology should consider the reaction kinetics, and should also be divided, as it is poorly written and full of unnecessary words, making it very difficult to follow. If the authors address those issues as well as the recommended English edition, I think that this article could be considered for publication in MDPY Polymers.
Reviewer 3 Report
Revision article: A Simple Approach to Fabricated Rice Husk 2 Reinforced Recycled HDPE composite foam and its 3 Mechanical Properties.
In my opinion, the article is not suitable for publication in its current form. The article needs to be significantly improved (to a significant extent).
Introduction is inconsistent, many sentences are unclear and incomprehensible. The abbreviations in Introduction (used for the first time) are not explained. Some explanations appear only in the research part, but earlier the reader does not understand them. In addition, appeared spaces between paragraphs in the text. They shouldn’t be there. In introduction the singular and plural are sometimes used wrong.
Examples of errors below:
Werse 33: Z pierwszego zdania nie wynika co oznacza :’’ fabricating high performance composite’’.
- What copmpsite?
Werse 34: What a sentence: ‘’ It is crucial to sustain nature by monitoring…’’ means? Werse 36: ‘’ natural environment and resources, while in the meantime innovating with new products that could minimise any negative environmental impact’’
- What innovation products do you mean? Please give some examples.
Werse 39: ‘’ in structural applications’’
- What structural applications do you mean? Give examples.
Werse 43: ‘’Recycled high density polypropylene (rHDPE)’’
– czy to oznaczenie jest zgodne z normÄ…? JeÅ›li tak proszÄ™ podać normÄ™.
- Does this research is according to the standard? If so, please provide/define this standard.
Werse 45: pure rHDPE
- rHDPE is a designation (symbol, symbol) for recycled HDPE or for pure HDPE? It is not known to this place.
Thus, the use of rHDPE not only minimises solid waste disposal, but also 44 reduces the final manufacturing cost.
- Sentence not clear. What final manufacturyng costs do you mean? What product costs do you mean?
Werse 46 usage of fibre ‘’
- What kind of fibre?
Werse 47: ‘’in fact, lighter 47 polymer composites with better strength and rigidity promote the biodegradable process and are 48 more economically competitive than the final composite material which could be produced’’
- In what way lighter polimer composites promote/favour degradation? Why this composities are more prone to degradation? Because they have a lower density? Please extend the issue.
In the next part of Introduction (werse 73) is a sentence ‘’ reduce the density and material costs’’. What is the relationship between reduction of costs light foam production and prone it to Degradation (susceptibility to degradation)?
Werse 55: ‘’ Surface modification of rice husk through chemical treatment such as alkali treatment, citric acid 56 treatment, anhydride, and permanganate, are commonly used to overcome this problem’’- Is such a treatment husk rice cost-effective? Is not doing it to expensive of production biodegradable HDPE? Werse 59: ‘’a great RH surface’’
- The abbreviation RH has not been explained (developed). What does it mean?
Werse 61: ‘’Some benefits of this 61 treatment compared to other chemical treatment..’’
- jakie sÄ… te inne chemical treatments o których mowa w tym zdaniu?
Werse 63: ozonolysis proces – chyba powinien być wspomniany wczeÅ›niej aby zaznajomić czytelnika o tym o jakÄ… metodÄ™ chodzi (chodzi o ozonolysis proces). Należy ten proces omówić szerzej. Werse 65: „The ozonolysis process also requires lower cost…’’
- The word 'requires' has rather been used incorrectly here. (maybe: causes, makes)
Werse 73: ‘’This fabrication of composite foam could reduce the density and material costs, and simultaneously increase the impact strength of the composite’’
- It is not clearly said what composite of foam is it? Composite are usulally made of two components. What components are used in this case?
Werse 74: Foam technology consists of a different series of processing methods to produce WPC parts, by creating large quantities of foam by foam agents, either by using physical or chemical foam agents, or by tension’’
-By tension? I think it was a different word.
Werse 87: RH
- the abbreviation RH has not been explained.
Werse: in Composite Preparation section no standard for UV/O3 system testing given. This standard is also not mentioned in the previous article of the authors [19]: Rajendran Royan, N.R.; Sulong, A.B.; Sahari, J.; Suherman, H. Effect of acid- and 397 ultraviolet/ozonolysis-treated MWCNTs on the electrical and mechanical properties of epoxy nanocomposites as bipolar plate applications. J. Nanomater. 2013, 2013, 1–8. Werse 109 ‘’…or the alkali treatment, RH was soaked in a 0.5M NaOH solution for 2 hours’’. Why exactly 2 hours? What was the reason to stop this process after 2 hours? Werse 100-116- Is this method of RH procesin possible to use in an industry or only in a laboratory extent? Werse 121: - In the tekst there are two kind of RH methods (untreated and treated). Powinno to być uwzglÄ™dnione w celu badaÅ„. It should be defined in aims (goals). Werse 131 : ,,Azocarbonamide (ADC) powder was used as an exothermic foaming agent’’
– Description of this foaming agent should be placed in Materials section. Tere was not given the name of a producer.
Werse 151: ‘’…were measured and calculated using formulae based on the morphology of the composite’’.
- Wchich formulaes? 1 and 2? You should complete.
Werse: Description of machnes and installations (e.g. extruder, presse and others) in w Composite Preparation section should be in Characterisation section. And Characterisation section should be before Composite Preparation section. Werse 188. You sould use a term ‘’Content of Cell per Area Unit’’ instead of ‘’cel density’’ Werse 188: In my opinion parameters in table 2 (e.g. 4.89 x 1011) are hight and width of cells, but not a cell density. Werse 205: Looking at the picture on fig.2 we can observe that shapes of cells are changed (not only diameters) (werse 186) Werse 328 Conclusions section: The authors didn’t give us the information how much they achieved their goals. There are not specific results of a research in Conclusions section. Werse 201 and 212 ‘’… with the optimum percentage (50 wt. % of RH)’’
50 wt. % of RH is the optimum? Not 10 wt.%?
Werse 311: There is no paragraph. Did you use some other additions e.g. anti-adhesive or processing additives except used by you coupling agent, and zinc oxide as an activator to reduce heat release
